# Research on Evaluation Index of Automotive Antenna Testing

**Meijun Qu** [1], **Siyang Sun** [2,*], **Huanhuan Jing** [2] and **Long Pei** [2]

1  School of Information and Communication Engineering, Communication University of China, Beijing 100024, China; qumeijun@cuc.edu.cn
2  China Telecommunication Technology Labs, China Academy of Information and Communications Technology, Beijing 100191, China; jinghuanhuan@caict.ac.cn (H.J.); peilong@caict.ac.cn (L.P.)
*  Correspondence: sunsiyang@caict.ac.cn; Tel.: +86-189-1001-0573

**Abstract:** Mounted locations and the ground plane structure have remarkable influences on the performance of roof-mounted automotive antennas. To distinguish this influence in radiation, figure of merits (FoMs), including total radiated power (*TRP*), near-horizontal part radiated power (*NHPRP*), and cumulative distribution function (*CDF*), are studied in this paper. It is proved that *TRP*s are almost the same with different mounting configurations. Because the radiation toward the horizon is a critical performance metric for automotive antennas, *NHPRP* is analyzed within certain degrees near the horizon. Even though a bigger deviation has been observed in *NHPRP*, the discrimination between different mounted scenarios is still not enough. Different from *TPR* and *NHPRP*, which are efficiency-based FoMs, *CDF* combines the gain values and the pattern shape together, achieving a comprehensive and intuitive insight into the antenna performance. It is more predictive and distinguishable in terms of the radiation pattern than *NHPRP* and *TRP*. Therefore, *CDF* can be utilized as a good supplement to existing metrics and can better distinguish the radiation performance of different antenna mounting configurations.

**Keywords:** automotive antenna; CDF; evaluation index; *NHPRP*; *TRP*





## 1. Introduction

The fifth-generation (5G) network has become a key driving force for the development of the internet of vehicles (IOV) because of its high speed, ultra-reliable, and low-latency characteristics [1–5]. In the meanwhile, vehicular communications play an important role in intelligent transportation systems. Modern cars are equipped with a large number of antennas working at different frequencies and covering many services, including radios (FM, HD), sensors, communication (3G, 4G, and 5G), Global Navigation Satellite System (GNSS), Wi-Fi, DSRC/V2X (Vehicle, Infrastructure, Cloud, etc.), anti-collision systems, etc. [6–10]. These antennas can be integrated at various locations inside and outside the vehicle, for example, on the roof, external mirror, front and rear bumper, or even in the dashboards [11–14]. There is still a lot to be explored when it comes to the evaluation of automotive antennas mounted at different positions on the vehicle.

Because the automotive antennas are highly integrated, the patterns and performance of antennas can be significantly affected by mounting locations and the shape of the vehicle body due to the strong coupling with the vehicle structure [15,16]. Since the radiation currents are spreading on the vehicle body, parts of the vehicle body close to the antenna must be considered as parts of the radiator when evaluating the performance of automotive antennas in real application scenarios. Hence, test results of the antenna module cannot represent the real performance in the system level of antennas on the vehicle. Because the full-vehicle automotive measurement system is quite expensive, few laboratories can afford such a large investment; international standard organizations, including the Cellular Telecommunications Industry Association (CTIA) and the 5G Automotive Association (5GAA), and operators are working on the standardization of automotive antenna ground

plane and considering how to test radiated performance for automotive antennas mounted on a ground plane going forward. The size/shape of the ground plane would have an impact on the measured pattern, which in turn would have an influence on the regulatory compliance of certain antenna systems. In theory, lower frequencies require a larger ground plane than higher frequencies. Concerns that making the ground plane too large will make it difficult to measure.

For many current vehicles (especially cars with model years 2005–2019), the main antenna configuration (GPS, FM, cellular, etc.) is generally centered on the back/center of the hood. If this becomes the default position for cellular antennas going forward, an off-centered mounting configuration on a round ground plane may provide a good compromise between accuracy and simplicity. On this basis, we compare the effects of various circular ground plane size and mounted locations (centered and off-centered) on the performance of automotive antennas mounted on it, so as to help researchers and operators better understanding its mechanism.

Moreover, accurate indication of the radiation pattern of the automotive antenna is another research hotspot. The impedance matching and radiation pattern will definitely be affected after the antenna mounting on the vehicle. For automotive antennas, radiation toward the horizon is a critical performance metric. However, different ground plane configurations could lead to different near horizon performances. Additionally, reflections off of the hood or trunk can yield a significant impact on performance near the horizon. Hence, the near horizon quantities have also been studied under different ground plane configurations in this paper.

Some organizations prefer to focus on evaluating the 3D Over-the-Air (OTA) quantities instead of trying to accurately assess the automotive antenna pattern or horizon quantities in real applications. However, these tests need a wireless radio module for automobiles and are models-dependent. Based on the existing generic test environment (where total radiated power (*TRP*) would be sufficient), we analyze the automotive antenna directly in this paper using a new evaluation index.

The gain and half-power beamwidth have been used as the primary metrics for evaluating the radiation performance of the traditional antenna. Nevertheless, they are less useful in evaluating the impact of mounting configuration on the performance of automotive antenna because of their inherent weakness, i.e., pattern shape is largely missing from these metrics. In many cases, cumulative distribution functions (CDFs) are commonly used in telecommunications for characterizing network performance. Here, a statistical approach based on CDFs is employed and allows designers to predict the ability to create or maintain an excepted radio link with greater probability. Note that CDF can be used as an index to evaluate the performance of automotive antennas and can distinguish the radiation performance of different antennas.

To sum up, we study the impact of circular ground plane size and mounted locations on the performance of the automotive antenna mounted on it. Then, to achieve an accurate indication of the installed radiation pattern and differentiate the various radiation performances, 3D radiation pattern, *TRP*, near-horizontal part radiated power (*NHPRP*), and CDF were analyzed step by step. Finally, it is found that CDF is an appropriate figure of merit (FoM) for operators and certification organizations when evaluating vehicle-mounted antennas for cellular communications.

## 2. Effect of Different Ground Configurations on Automotive Antennas

For IOV communications, near horizon quantities are critical, based on the channel model of vehicle to road (V2R) and vehicle to infrastructure (V2I) scenarios, where the base station antennas are located in elevation, and vehicle to vehicle (V2V) scenario in which communications occurred between vehicles in the horizon. Monopole-like antennas, including shark fin antennas and whip antennas, which radiate uniformly toward the horizon, are the most common form of roof-mounted automotive antennas. However, when the antenna is mounted on the roof, it cannot keep uniform radiation on the horizontal

because of the strong coupling with vehicle structures. In order to study this change in radiation properties, we simulate the impact of various ground plane configurations on the performance of an automotive antenna mounted on it in this section. Note that for simplicity, a monopole antenna is adopted in simulations.

As a reference, we first consider the monopole located in the center of a circular ground plane. The monopole has a height of 0.22 $\lambda_0$ at 2 GHz, while $\lambda_0$ is the resonant wavelength in free space and the radius of the ground plane is set to 2 $\lambda_0$. The current distribution on the ground plane is shown in Figure 1. The maximum current magnitude on the ground plane appears at the feeding point and reduces rapidly away from that point. At the distance of 0.25 $\lambda_0$ and 0.5 $\lambda_0$ from the center, the current magnitude reduces to around −30 dB and −45 dB of its maximum, respectively.

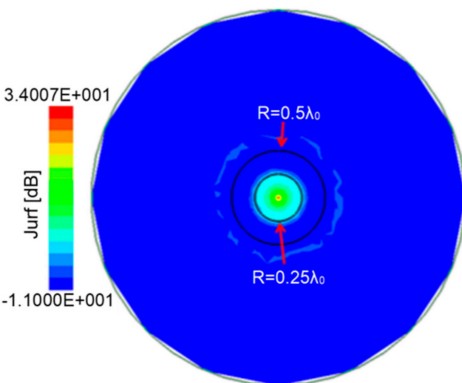

**Figure 1.** The current distribution for monopole antenna centered on the ground plane.

The comparison of the normalized impedance of the antenna between different ground plane sizes is demonstrated in Figure 2. It can be seen that the impedance changes slightly over the entire frequency range when the ground radius increases from 0.5 $\lambda_0$ to 10 $\lambda_0$, especially for a radius larger than 1 $\lambda_0$. This can be explained as the majority of the current is distributed in the regions indicated in Figure 1, out of which the current is extremely weak and has little impact on the antenna's impedance and bandwidth properties. Namely, the impact of the ground plane on antenna impedance properties can be ignored when its radius larger than 0.5 $\lambda_0$.

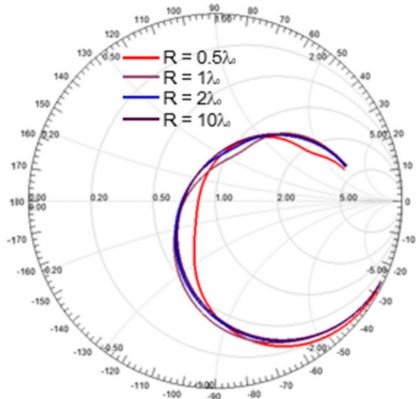

**Figure 2.** Comparison of normalized impedance between different ground plane sizes (R = 0.5 $\lambda_0$, 1 $\lambda_0$, 2 $\lambda_0$, and 10 $\lambda_0$).

Radiation patterns of the center located monopole on different sizes of the ground plane are shown in Figure 3. Considering the symmetry, gain values at phi = 90° in azimuth are collected and compared. As expected, the peak directions are elevated above the finite ground plane's horizon. Taking 15 $\lambda_0$ as a reference, when the radius is larger than 2 $\lambda_0$,

the discrepancy between peak gains is less than 2 dB. Compared to impedance properties, radiation patterns of the center-located monopole antenna are more vulnerable to the ground plane sizes.

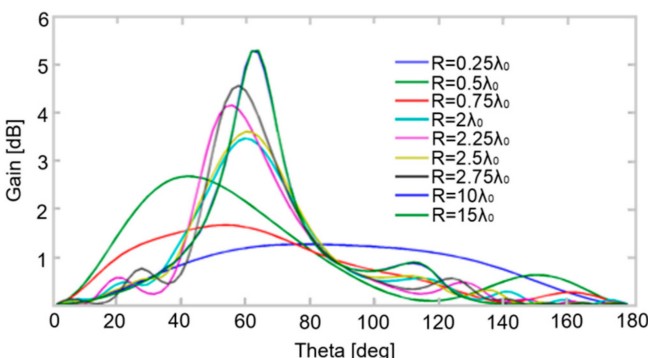

**Figure 3.** Radiation patterns of center located monopole on different sizes of the ground plane.

Placing the antenna on the side of the ground plane may be one effective way to mimic the effect that is observed when an antenna is mounted on the back section of the roof. Hence, monopole antennas in several off-centered mounting configurations are also simulated and analyzed in addition to the centered-mounted scenario. When the monopole height and the radius of the ground plane are set to $0.22\,\lambda_0$ and $2\,\lambda_0$, respectively, the current distributions on the ground plane in centered and off-centered mounting configurations are shown in Figure 4.

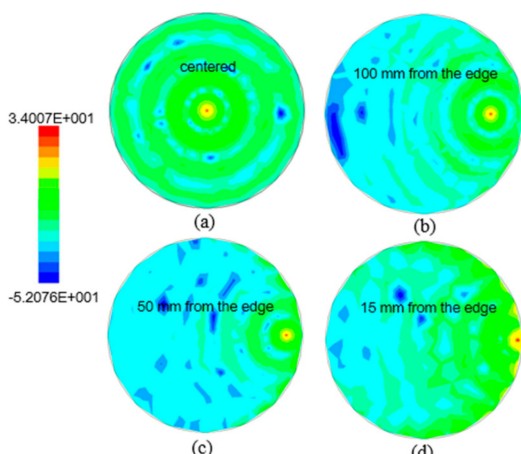

**Figure 4.** The current distributions on the ground plane in centered and off-centered mounting configurations—(**a**) centered, (**b**) 100 mm, (**c**) 50 mm, and (**d**) 15 mm from the right edge.

As the monopole moves toward the right edge, the symmetrical annular current distribution changes gradually. From Figure 4a–c, the current distribution in the region indicated in Figure 1 where the majority of current changes slightly, thus the impedance and bandwidth properties change indistinctly. In Figure 4d, when the feed port approaching the right edge of the ground plane, the regions aforementioned are cut by the edge, part of the current propagates along the edge, similar to the standing wave. The closer the distance from the edge, the more obvious this phenomenon. This standing wave distribution of current along the edge of the ground plane visibly changes the impedance properties of the monopole with respect to those exhibited in the center-mounted scenario. The comparison of normalized impedance between centered and off-centered mounting configurations is shown in Figure 5.

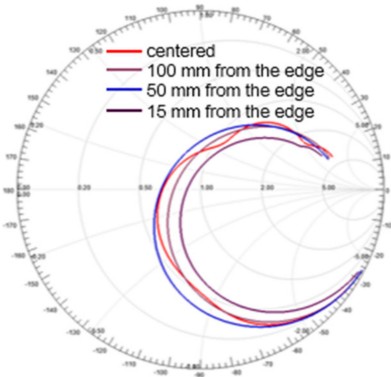

**Figure 5.** Comparison of normalized impedance between centered and off-centered mounting configurations.

From Figure 5, as the monopole shifts from the center to the right edge of the ground plane, the resonant frequency shifts to the upper frequency slightly while the radiation impedance increases. Compared to the impedance property, a more significant change in radiation patterns can be observed, as illustrated in Figure 6.

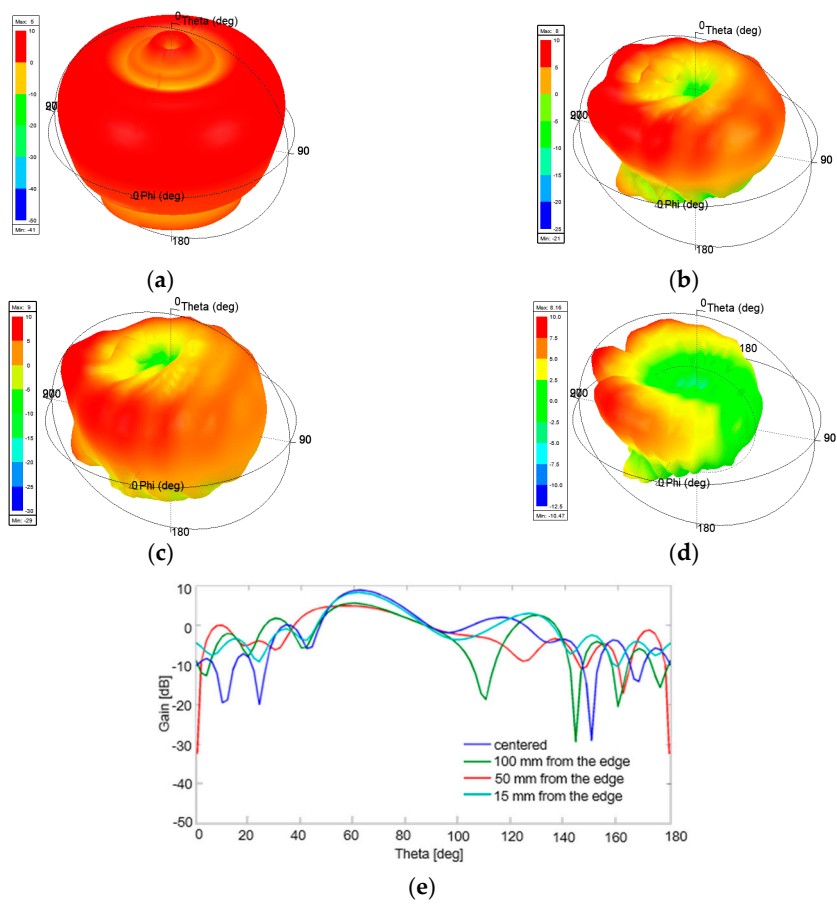

**Figure 6.** Comparison of antenna radiation pattern between centered and off-centered mounting configurations, (**a**) centered, (**b**) 100 mm, (**c**) 50 mm, (**d**) 15 mm from the right edge, and (**e**) cut of the main beam.

As expected, the pattern shape changes dramatically when the monopole approaches the ground edge from Figure 6a–d. The symmetry of the pattern is destroyed as the monopole moves to the right edge. Meanwhile, the depth of the null in axial directions, which is a fundamental behavior of a typical monopole radiation pattern, decreases in



Figure 6e. This alternation can be explained as the destruction of symmetry of current distribution on the ground plane and the radiation of current flowed along the ground edge. However, for off-centered configurations, if the regions indicated in Figure 1 are not destroyed, i.e., the majority of the current is not disturbed, the main beam portion of the pattern is almost the same.

Metrics such as peak gain, half-power beamwidth, phase center, etc., have been used as the primary metrics for evaluating radiating performance. Nevertheless, they do not reflect the comprehensive performance of automotive antennas to some extent.

### 3. *TRP* as the Evaluation Index

*TRP* is the average of spherical effective isotropic radiated power, which is measured by sampling the effective isotropic radiated powers (*EIRP*s) of two polarizations at various locations surrounding the device, as can be seen in Figure 7. *TRP* has been used as the primary metric for evaluating radiating performance in CTIA test plan for many years. For a complete sphere measured with *N* theta intervals and *M* phi intervals, both with even angular spacing, the *TRP* is calculated as follows [17]:

$$TRP \cong \frac{\pi}{2NM} \sum_{i=1}^{N-1} \sum_{j=0}^{M-1} \left[ EiRP_\theta(\theta_i, \phi_j) + EiRP_\phi(\theta_i, \phi_j) \right] \sin(\theta_i) \tag{1}$$

where $EiRP_\theta (\theta_i, \phi_j)$ and $EiRP_\phi (\theta_i, \phi_j)$ are *EIRP* values measured in theta and phi polarization at the location of $(\theta_i, \phi_j)$, respectively. The *TRP*s could be derived from the simulated radiation pattern. These *TRP*s of all antenna mounting configurations listed in Table 1 are calculated based on Equation (1).

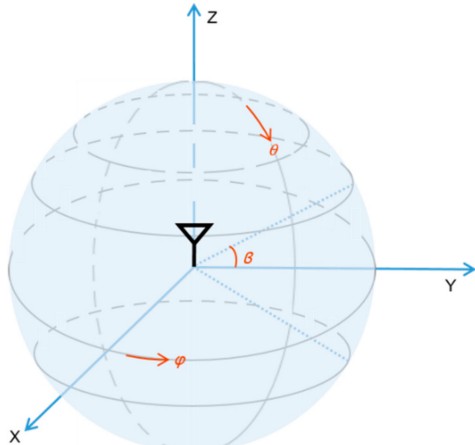

**Figure 7.** The test schematic of total radiated power (*TRP*).

As shown in Figure 8, for different mounted locations in the off-centered scenario and different sizes of the ground plane in the centered scenario, the calculated *TRP*s are almost the same. For simplicity, the input powers in all simulations throughout this paper are set to 0 dBm. The delta between the maximum and minimum is around 0.9 dB. This is because *TRP* is an efficiency-based FoM, and the impedance mismatch is the dominant factor affecting radiation efficiency. In contrast, the changes in impedance property are slight and the reflection coefficients at the resonant/interested frequency are lower than −10 dB, thus *TRP*s are similar and deltas between them are small, which suggests that *TRP* are FoM insensitive to mounted locations and ground sizes. Therefore, *TRP* cannot be the only FoM for the evaluation of automotive antenna. Relying only on integral values such as *TRP* without considering pattern data is strongly discouraged.

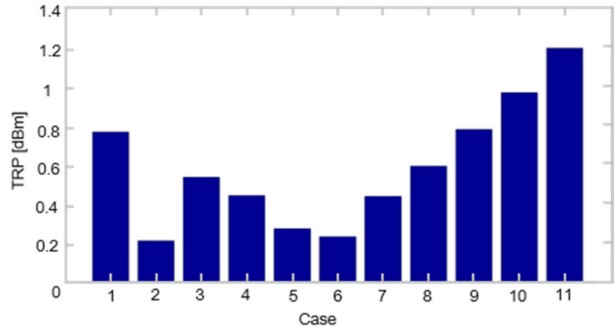

**Figure 8.** Comparison of *TRP*s between different antenna mounting configurations.

**Table 1.** Antenna mounting configurations.

| Case | Configuration | |
| --- | --- | --- |
| Case 1 | R = 2 $\lambda_0$ | 15 mm from the edge |
| Case 2 | R = 2 $\lambda_0$ | 50 mm from the edge |
| Case 3 | R = 2 $\lambda_0$ | 100 mm from the edge |
| Case 4 | R = 2 $\lambda_0$ | centered |
| Case 5 | R = 3 $\lambda_0$ | 15 mm from the edge |
| Case 6 | R = 3 $\lambda_0$ | 50 mm from the edge |
| Case 7 | R = 3 $\lambda_0$ | 100 mm from the edge |
| Case 8 | R = 3 $\lambda_0$ | 150 mm from the edge |
| Case 9 | R = 3 $\lambda_0$ | 200 mm from the edge |
| Case 10 | R = 3 $\lambda_0$ | 300 mm from the edge |
| Case 11 | R = 3 $\lambda_0$ | centered |

## 4. NHPRP as the Evaluation Index

For automotive applications, radiation toward the horizon and related performance metrics are of great importance. Hence, we calculate the *NHPRP* data within certain degrees near the horizon in this section. The test schematic is shown in Figure 9.

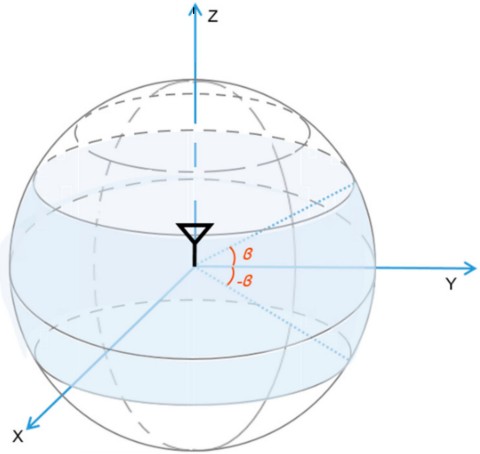

**Figure 9.** The test schematic of near-horizontal part radiated power (*NHPRP*).

For a complete sphere measured with $N = 12$ theta intervals and M phi intervals, both with even angular spacing, for instance, the power radiated over $\pm 45°$ near the horizon can be calculated as follows [17]:

$$NHPRP_{\pm 45°} \cong \frac{\pi}{2NM} \left( \frac{cut_3 + cut_9}{2} + \sum_{i=4}^{8} cut_i \right) \tag{2}$$

where

$$cut_i = \sum_{j=0}^{M-1} \left[ EiRP_\theta \left( \theta_i, \phi_j \right) + EiRP_\phi \left( \theta_i, \phi_j \right) \right] \sin(\theta_i) \tag{3}$$

$NHPRP_{\pm 30°}$ takes into account angles within $\pm 30°$ from the horizon, while $NHPRP_{\pm 15°}$ takes into account angles $\pm 15°$ within from the horizon. The calculation formulas of $NHPRP_{\pm 30°}$, $NHPRP_{\pm 15°}$ are similar to Equations (2) and (3).

The *NHPRP*s for all antenna mounting configurations listed in Table 1 are calculated, the comparison of which is summarized and tabulated in Table 2. The $NHPRP_{\pm 45°}$, $NHPRP_{\pm 30°}$, and $NHPRP_{\pm 15°}$ for different mounting configurations are shown in Figure 10, respectively. It is found that the delta between different mounted scenarios is larger than that of *TRP*s but still small from the value range of the vertical coordinates. Therefore, *NHPRP* has better prediction and discrimination than *TRP* and can be used as a supplement to *TRP* in evaluating the performance of the automotive antenna. Moreover, $NHPRP_{\pm 15°}$ has the highest discrimination among all indexes.

**Table 2.** Summarization of *NHPRP*s comparison.

| Case | $NHPRP_{\pm 45°}$ (dBm) | $NHPRP_{\pm 30°}$ (dBm) | $NHPRP_{\pm 15°}$ (dBm) |
|---|---|---|---|
| Delta for all | 0.9 | 1.18 | 1.52 |
| Delta for R = 2 $\lambda_0$ | 0.6 | 0.83 | 1.30 |
| Delta for R = 3 $\lambda_0$ | 0.68 | 0.73 | 1.20 |

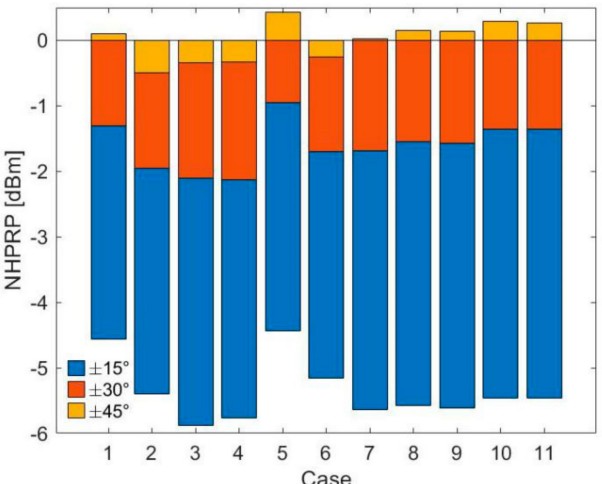

**Figure 10.** Comparison for different mounted locations and ground sizes ($NHPRP_{\pm 15°}$, $NHPRP_{\pm 30°}$, and $NHPRP_{\pm 45°}$).

*NHPRP* relies on the 3D radiation pattern taken from angels within certain degrees near the horizon to create a partial radiated power. This is an improvement over *TRP* because it takes into account the most heavily angles toward which communications in V2X may occur. Nevertheless, it has some inherent weakness in that the average power is considered, and pattern shape is largely missing from this FoM.

## 5. CDF as the Evaluation Index

*TRP* is an efficiency-based FoM, which results in the absence of pattern data. As an improvement from *TRP*, *NHPRP* quantities are simply a portion of the *TRP*, which is still based on efficiency or average.

CDF is the integral of the probability density function, which can completely describe the probability distribution of a real random variable. CDFs are commonly used in telecommunications for characterizing the performance of networks. In this paper, *EIRP* values over a subset of sampling points are collected for CDF analysis, which can provide an indication of the likelihood of having a certain *EIRP* value in a random direction. In other words, a statistical approach based on CDFs may offer a comprehensive insight into the radiation performance and allow the ones to predict system performance with greater probability.

The CDF curves of radiation for different mounted locations and ground sizes within different ranges in elevation are depicted in Figures 11–13, respectively. The antenna mounting configurations refer to Table 1. The CDFs comparison is summarized and tabulated in Table 3.

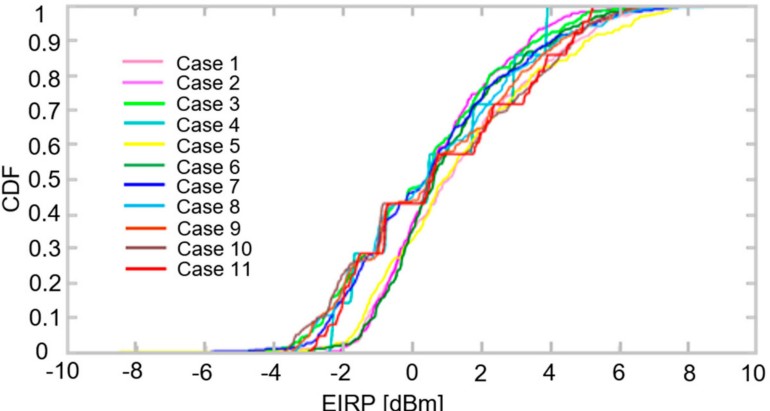

**Figure 11.** Effective isotropic radiated power (*EIRP*) of cumulative distribution functions (CDFs) comparison within 70° to 100°.

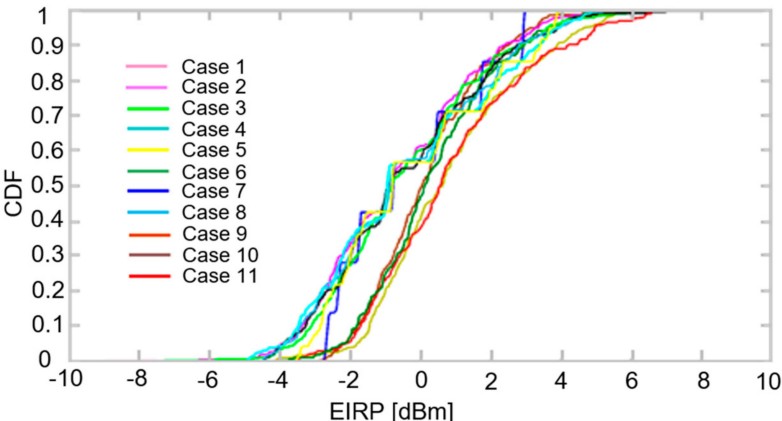

**Figure 12.** *EIRP* CDFs comparison within 75° to 105°.

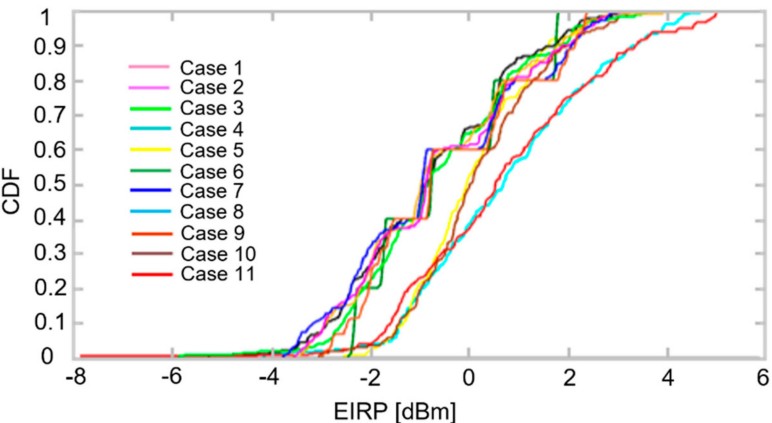

**Figure 13.** EIRP CDFs comparison within 80° to 100°.

**Table 3.** Summarization of CDFs comparison.

| Elevation | Delta $EIRP_{CDF-80\%}$ (dBm) | | | Delta $EIRP_{CDF-60\%}$ (dBm) | | |
|---|---|---|---|---|---|---|
| | All | R = 2 $\lambda_0$ | R = 3 $\lambda_0$ | All | R = 2 $\lambda_0$ | R = 3 $\lambda_0$ |
| 80–100° | 1.98 | 1.98 | 1.75 | 2.18 | 1.93 | 1.96 |
| 75–105° | 1.51 | 1.33 | 1.27 | 1.28 | 1.22 | 1.20 |
| 70–100° | 1.48 | 1.29 | 1.05 | 1.08 | 0.95 | 0.8 |

$EIRP_{CDF-X\%}$ indicates an X% probability of having a certain *EIRP* value in a random direction within the elevation considered. It can be seen from Table 3 that the maximum radiation performance variation can be achieved under the *EIRP* outage level of 60% within 80° to 100° in elevation.

Compared to *TRP*, *NHPRP*s and CDFs show larger variations between different mounted scenarios, suggesting they are more suitable for characterizing differences in automotive antenna performance. It is clear from Tables 2 and 3 that CDFs are more distinguishable than *NHPRP*s. This is because *NHPRP*s are average values where the difference in the radiation pattern is concealed, while CDFs combine gain/*EIRP* and the pattern shape over given elevation angles together. Therefore, $EIRP_{CDF-60\%}$ is recommended as the FoM to differentiate and rank automotive antenna performance. Designers could also specify an acceptable value based on a percentile threshold.

## 6. Conclusions

This paper analyzes the impact of mounted locations and ground sizes on the radiation performance of automotive antennas. Afterward, to accurately indicate the mounted radiation performance of automotive antennas, *TRP*, *NHPRP*, and CDF are studied individually. Because the *TRP* is the FoM directly determined by efficiency, it is insensitive to the mounted locations and ground sizes of the automotive antenna. *NHPRP* takes the radiation toward the horizon as a critical performance metric and has better prediction and discrimination than *TRP*. Nevertheless, it is not high enough to distinguish between different antenna performances. *TRP* and *NHPRP* can only provide average radiation performance, and the pattern shape is largely missing from these FoMs. The proposed metric CDF combines the gain values and the pattern shape together, providing a single FoM, which offers comprehensive and intuitive insight into the antenna performance. Therefore, it is believed that CDF analysis could become a good supplement to existing metrics and be used in system link budget analysis. Meanwhile, *EIRP* CDF-60% is recommended as the FoM to differentiate and rank automotive antenna performance.

**Author Contributions:** Conceptualization, S.S.; methodology, M.Q.; investigation, H.J. and L.P.; writing—original draft preparation, M.Q.; writing—review and editing, S.S.; visualization, M.Q. All authors have read and agreed to the published version of the manuscript.

**Funding:** This research was funded by the Key Research and Development Project of Guangdong Province, grant number 2020B0101080001 and the Fundamental Research Funds for the Central Universities.

**Data Availability Statement:** Data sharing not applicable.

**Conflicts of Interest:** The authors declare no conflict of interest.

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
