# Peer review of "Research on Evaluation Index of Automotive Antenna Testing"

_electronics, doi:10.3390/electronics10040515_

Round 1

Reviewer 1 Report

Line 55: understanding what? 

Figures 10, 11, and 12 could be plotted together to facilitate comparison.

Figure 13, 14, and 15, it is better to use a legend that illustrates the differences among cases.

Reviewer 2 Report

Telecommunications development trends are such that wireless technologies are becoming an integral part of almost any engineering system. One of the most widespread engineering systems is road transport, which has become an integral part of modern life. Development trends such as intelligent transport systems, self-driving cars place even greater demands on the exchange of information between road users.

Therefore, the topic of this work is very relevant. The authors decide the issues of choosing the most appropriate place for placing the radio antenna on the car. The solution to this problem is aimed at increasing the efficiency of the applied wireless communication system and allows a positive effect on the functioning of both man-to-man communication systems and machine-machine systems.

In their work, the authors investigate the effect of the choice of antenna placement on the antenna impedance and directivity. These are important characteristics.

The authors have carried out a fairly large amount of research and obtained interesting and useful results.

However, for a better presentation, the material should be well checked and slightly modified.

Unfortunately, the authors made a lot of mistakes in the design of illustrations and mathematical expressions. All these shortcomings are completely removable and do not affect the main ideas of the article.

However, there are quite a few inaccuracies and they must be eliminated before publishing the material!

Disadvantages:

  1. In Figure 2 you need to bring the units of measurement.
  2. The authors call the wavelength λ0 in the text, but the figures show the designations without the “0” index, ie just λ. The same designations for the same values should be used in the text and in the figures (Figures 1, 2 and 3).
  3. For Figure 3, you need to explain what exactly is shown in the figure. The vertical axis of the graph shows the values that are designated as Gain, it is necessary to explain in the text what these values reflect (for example, gain or attenuation) from the text, it is not clear.
  4. Figure 4 is wrong!

It completely repeats Figure 3. The text contains references to Figures 4 a, b, c and d. There are no such drawings.

A fix is required.

  1. Formula (1) must be described in the text. There are two EiRP functions in the formula, you need to explain what this value is (Effective Isotropic Radiated Power). When describing the formula in the text, it is better to give those symbols that are used in the formula, and for their transcription in English “phi”, “theta”. This gives an unambiguous understanding of the above expressions and avoids errors.
  2. It is possible that formula (2) is written with an error! In total, the i index should vary from 4 to 8, according to [17], for the authors from 1 to 8.

Correction or clarification required.

It is also required to clarify all the symbols used in the formula. This must be done for all formulas given.

  1. It is necessary to check formula (3), it differs from [17], or explain what caused the differences.

(The original formula uses Mi and divides the expression by Mi.)

  1. The graphs of Figures 8, 10-12 show the values in dBm, it is necessary to explain in the text what was the level of the radiation power (generator Tx power level dBm).
  2. In Figures 13,14,15, you need to indicate the value and units of measurement on the horizontal axis, probably it should be NHPRP, dBm.
  3. Conclusions should be expanded. The authors suggest using of CDF as an indicator of antenna quality. This is probably the case. But the authors should describe in more detail how this indicator characterizes the antenna. What distributions are desirable, what should you strive for.

I believe that the authors need to eliminate the noted shortcomings!

I also advise the authors to check all the work very carefully.

Round 2

Reviewer 1 Report

It's better to have this article revised by an English native speaker. He/She could provide useful advice on your language and style of presentation.